# Normalization of the ATP1A1 Signalosome Rescinds Epigenetic Modifications and Induces Cell Autophagy in Hepatocellular Carcinoma [note 1]

**DOI:** 10.3390/cells12192367

**Published:** 2023-09-27

**Authors:** Pradeep Kumar Rajan, Utibe-Abasi S. Udoh, Yuto Nakafuku, Sandrine V. Pierre, Juan Sanabria

**Affiliations:** 1Department of Surgery, Marshall Institute for Interdisciplinary Research, Marshall University School of Medicine, Huntington, WV 25701, USA; rajan@marshall.edu (P.K.R.); udohu@marshall.edu (U.-A.S.U.); nakafuku@marshall.edu (Y.N.); pierres@marshall.edu (S.V.P.); 2Department of Nutrition and Metabolomic Core Facility, Case Western Reserve University School of Medicine, Cleveland, OH 44100, USA

**Keywords:** ATP1A1, hepatocellular carcinoma (HCC), MASH, epigenetic changes, autophagy

## Abstract

Hepatocellular carcinoma (HCC) is the third leading cause of cancer-related death worldwide. In metabolic dysfunction-associated steatohepatitis (MASH)-related HCC, cellular redox imbalance from metabolic disturbances leads to dysregulation of the α1-subunit of the Na/K-ATPase (ATP1A1) signalosome. We have recently reported that the normalization of this pathway exhibited tumor suppressor activity in MASH-HCC. We hypothesized that dysregulated signaling from the ATP1A1, mediated by cellular metabolic stress, promotes aberrant epigenetic modifications including abnormal post-translational histone modifications and dysfunctional autophagic activity, leading to HCC development and progression. Increased H3K9 acetylation (H3K9ac) and H3K9 tri-methylation (H3K9me3) were observed in human HCC cell lines, HCC-xenograft and MASH-HCC mouse models, and epigenetic changes were associated with decreased cell autophagy in HCC cell lines. Inhibition of the pro-autophagic transcription factor FoxO1 was associated with elevated protein carbonylation and decreased levels of reduced glutathione (GSH). In contrast, normalization of the ATP1A1 signaling significantly decreased H3K9ac and H3K9me3, in vitro and in vivo, with concomitant nuclear localization of FoxO1, heightening cell autophagy and cancer-cell apoptotic activities in treated HCC cell lines. Our results showed the critical role of the ATP1A1 signalosome in HCC development and progression through epigenetic modifications and impaired cell autophagy activity, highlighting the importance of the ATP1A1 pathway as a potential therapeutic target for HCC.

## 1. Introduction

Hepatocellular carcinoma (HCC) is the dominant type of liver cancer, representing 75–85% of primary organ malignancies. Epidemiological studies have shown that it is the sixth leading cause of cancer death in Western countries [1,2] and the second most common cause of cancer mortality in East Asia and sub-Saharan Africa [1,3,4]. Metabolic dysfunction-associated steatohepatitis (MASH) is a major risk factor for the development of HCC, with a variety of biological behaviors and multiple abnormal phenotypes driven by genetic and epigenetic modifications [5]. Current advances in tumor biology and molecular genetic profiling include cell-signaling pathways and molecular mechanisms involved in the pathogenesis of HCC, providing opportunity to identify novel targets for therapeutic development.

Genetic programming is directed by epigenetic information that further translates into gene expression from cell-specific developmental and metabolism-related changes [6,7,8]. Malignant transcriptional re-programming is mediated, at least in part, through the activation of proto-oncogenes and inactivation of tumor-suppressor genes, often leading to the widespread dysregulation of gene expression profiles and disruption of signaling networks that control cell differentiation, proliferation and thus cellular functions [9,10,11]. The liver epigenome is sensitive to variable environments caused by circadian cues, metabolic instabilities, and external stimuli [12]. Conditions such as diabetes, obesity, hepatitis B virus/hepatitis C virus (HBV/HCV), alcohol-induced cirrhosis, and smoking [13] among other factors trigger the induction of reactive oxygen intermediates (ROI), promoting cell-oxidative stress, dysregulation of the cell autophagy activity, and chronic inflammation, which in turn enhance the dysregulation of genetic and epigenetic changes in the liver [14,15]. Abnormal cell autophagy has been implicated not only in HCC development but in tumor progression, including proliferative growth and metastasizing [16,17].

Cellular-redox imbalance and chronic inflammation are also observed when the α1-subunit of Na/K-ATPase (ATP1A1) signaling is deregulated, as reported in various pathological conditions [18,19,20,21,22]. Our previous work has shown that normalization of the ATP1A1 signalosome attenuates MASH progression by interrupting an ROI amplification loop through reactivation of the mitochondrial metabolism that also promoted an apoptotic switch-off [23]. Furthermore, we proposed that the normalization of ATP1A1 signaling restored its tumor suppressor role through a cellular apoptotic switch-on mediated by a balance on the Survivin–SMAC proteins [23]. We further hypothesize that a dysregulation of ATP1A1 signaling in the liver promotes epigenetic reprogramming and drives HCC development. In addition, we studied the effect of ATP1A1 signalosome normalization on post-translational histone modifications and cell autophagy and their impact in HCC development and progression.

## 2. Materials and Methods

### 2.1. Cell Models

Human HCC cell lines Hep3B (HB-8064) and SNU475 (CRL-2236) from ATCC (Manassas, VA, USA), free of mycoplasma contamination (RT-PCR kit, from MycoAlert PLUS Mycoplasma Detection Kit, Lonza, Rockland, MD, USA), underwent 2-5 passages prior to experimental use. Hep3B cells were grown in culture with high-glucose DMEM media supplemented with 10% FBS and 1% Penicillin/Streptomycin. SNU475 cells were cultured in RPMI 1640 media supplemented with 10% heat inactivated FBS and 1% Penicillin/Streptomycin. All cells were maintained in a 37 °C humidified incubator in the presence of CO_2_ at 5%. Cryopreserved primary human hepatocytes were obtained from Life Net Health (Virginia Beach, VA, USA) and cultured according to the supplier’s instructions using LifeNet Human Hepatocytes Culture and Supplement media (MED-HHCM& MED-HHPMS). Cells in culture were either untreated or treated with pNaKtide as previously described (IC_50_ of pNaKtide for Hep3B = 62.5 µM, and for SNU475 = 6 µM) [23]. At 4 h or 18 h post-treatment, cells were harvested for histone extraction and the subsequent analysis of modified histone levels.

### 2.2. Mice Models

NOD SCID mouse is an immune-deficient mouse model lacking functional T and B cells. It is low on the leakiness scale, making it suitable for studies involving malignant xenograft cells [24]. Seven-week-old female NOD SCID mice (NOD.CB17-Prkdcscid/J, JAX stock #001303, Jackson Laboratory, Farmington, CT, USA) were used in a tumor xenograft model of HCC. Human Hep3B cells, transfected with dual-luciferase (firefly luciferase) and fluorescent (mCherry) lentiviral constructs (SHP001, Sigma, St. Louis, MO, USA), were implanted subcutaneously (10^6^ cells) into each animal’s flank/back. The mice were randomly divided into two groups after 7 days of cell implantation. *Group 1*: NOD SCID mice implanted with human Hep3B cells and treated IP with 0.9% normal saline (NS). *Group 2*: NOD SCID mice implanted with human Hep3B cells and treated IP with pNaKtide in NS (30 mg/kg. BW, three times per week). Tumor luminescence was analyzed using the IVIS system (IVIS lumina XRMS Series III, PerkinElmer Inc, Waltham, MA, USA) and video-monitored weekly [25]. Animals were euthanized after eight weeks or when the tumor reached 1500 mm^3^, whichever came first (to ensure humane endpoint according to IACUC protocol; IACUC #: 741, 5 October 2020—[1565850] Src-phosphorylation role in HCC tumor reduction in the NOD-SCID mice). Tumor and liver tissues were collected and stored at −80 °C for future analysis.

*The MASH-HCC mouse model* (STAM™ mice, Tokyo, Japan) was produced by injecting Streptozotocin (200 µg STZ, Sigma, MO, USA) to neonatal male C57BL/6J mice 2 days after birth. The injected animals were exposed to a high-fat diet (HFD32, CLEA Japan, Inc., Tokyo, Japan) ad libitum after 4 weeks of age. The mice developed MASH at 12 weeks with the presence of HCC. Mice fed a normal mouse chow diet (NMC, Bio-Serv, Flemington, NJ, USA) served as the control. After 12 weeks, mice were randomly divided into the control and treatment group (*n* = 7 per group), and the study continued for an additional 12 to 16 weeks. For early-stage MASH-HCC, the experimental duration was 24 weeks (12 W of HFD and 12 W of treatment), whereas for late-stage MASH-HCC (16 W of HFD and 12 W of treatment), the total experimental duration was 28 weeks (see animal flow chart of our previous study) [23]. *Group 1*: NMC with no treatment; *Group 2*: HFD with no treatment; *Group 3*: HFD treated with low dose of pNaKtide in NS IP (2 mg/kg TBWX3 a week); *Group 4*: HFD treated with high dose of pNaKtide solutions were (10 mg/kg TBWX3 a week) prepared on the day of treatment. Livers were collected and washed with 0.9% normal saline at RT to be divided into two parts for being snap-frozen in liquid nitrogen and stored at −80 °C or fixed at 4 °C in 10% formaldehyde. All animal studies were approved by the University Institutional Animal Care and Use Committee (IACUC # 662, 14 October 2016—MASH mouse model study) in accordance with the National Institutes of Health (NIH) Guide for Care and Use of Laboratory Animals.

pNaKtide is a 33 synthetic amino-acid peptide (GRKKRRQRRRPPQSATWLALSRIAGL CNRAVFQ; MW 3877.64) made up of 20 amino acids sequence (NaKtide) that mimics human ATP1A1 sequence critical for Src regulation and downstream signaling [26,27] preceded by 13 amino acids of the TAT leader sequence. The TAT leader sequence is a cell-penetrating sequence that aids large molecules to cross the cell membrane. Thus, pNaKtide distributes not only extra and intracellularly but also resides at the cell membrane, whereby blocking Src phosphorylation normalizes the ATP1A1/Src signalosome [26,27,28]. Previous studies showed that under physiological conditions, pNaKtide is readily taken up in vivo by the heart, kidney, liver and fat tissues, showing a plasma and cell membrane distribution [28].

### 2.3. Immunofluorescence Staining

Immunofluorescence studies in human normal hepatocytes and human HCC cell lines were performed as previously described [26]. In brief, cells were plated on glass coverslips and allowed to reach 70% confluency. Then, they were fixed with ice-cold methanol (10 min), permeabilized (0.05% Triton X-100) and incubated with a monoclonal or polyclonal antibody overnight (Appendix A). The next day, slides were incubated with secondary antibodies and mounted on slides with Vectashield mounting media containing DAPI (Vector Laboratories, Inc., Newark, CA, USA, H-1800). Images were taken at 40× or 63× magnification, with a zoom factor of 2.5/5.0, using a confocal microscope (Leica TCS SP5 II).

Paraffin-embedded hepatic and tumor tissue sections were dewaxed, heated in water bath in citrate buffer (pH 6.0) at 97 °C for 20 min, cooled to RT (20 min), rinsed with deionized water, and equilibrated in PBST (PBS + 0.1% TritonX100, pH 7.4) with gentle agitation (2 × 5 min at RT). After blocking tissues for non-specific binding sites (10% normal goat serum in BSA-1%/PBS, pH 7.4 for 1 h at RT), they were incubated with primary antibodies overnight at 4 °C. Binding was visualized after the addition of fluorescent secondary antibodies in 1% BSA/PBS for 1 h at RT (Appendix A). Finally, sections were washed in PBS, mounted in Vectashield antifade mounting medium with DAPI (Vector Laboratories, Inc., H-1800) and observed under Leica confocal microscope (TCS SP5 II). The images were taken at 40× and 63× magnification.

### 2.4. Histone Extraction and Quantitation of Histone H3-acetyl-K9 (H3K9ac) and Histone H3-tri-methyl-K9 (H3K9me3)

Total histone proteins were extracted from Hep3B cells, SNU475 cells and tumor tissue using a histone extraction kit following the manufacturer’s protocol (Abcam, Boston, MA, USA, ab113476, IL). Briefly, cells were harvested and pelleted by centrifugation at 1000× *g* rpm for 5 min at 4 °C. The tissue was homogenized using pre-lysis buffer (1:1; vol:vol) and centrifuged at 3000× *g* rpm for 5 min at 4 °C. The pellets were re-suspended in lysis buffer (1:3; vol:vol) and incubated on ice for 30 min, which was followed by centrifugation at 12,000× *g* rpm for 5 min at 4 °C. The supernatant (containing acid-soluble proteins) was transferred into a fresh vial. Dithiothreitol buffer (1:0.3; vol:vol) was added to the supernatant and stored at −80 °C. Protein concentration was accessed using the Lowry method and bovine serum albumin (BSA) as standard. Histone H3 acetyl and tri-methyl K9 were measured by Enzyme Linked Immuno-Sorbent Assay (ELISA) (Abcam, ab115104; ab115064, IL) according to the manufacturer’s instructions.

### 2.5. Autophagy Vacuoles

Immunofluorescence staining for autophagy vacuoles was conducted using an autophagy detection kit (Abcam, ab139484; Boston, MA, USA). Cells were plated on a cover slip, placed into 6-well plates and incubated at 37 °C for 1 day. After 3XPBS washes, cells were fixed (paraformaldehyde 4% for 15 min) and washed 3 times in PBS. After blocking with 1% BSA (A3294, Sigma-Aldrich; Merck, Germany) for 30 min at RT, the fluorescent dyes for nuclei staining and autophagy detection were added and incubated for 30 min at RT. After three washes with PBS, the slides were observed using a Leica confocal microscope (TCS SP5 II), and images from five non-overlapping fields per slides from each of five independent experiments were captured at 63× magnification with a zoom factor of 2.5. The number of autophagic vacuoles (green fluorescence) was counted by ImageJ software (NIH, Bethesda, MD, USA) and normalized by the area to give the number of vacuoles/unit area.

### 2.6. LC3-II, an Autophagy Marker

LC3-II protein was quantitated using an Autophagy ELISA kit (MBS169564, SanDiego, CA, USA). A selective permeabilization procedure was used to remove the cytosolic pro-LC3 and LC3-1 while retaining the autophagosome-bound LC3-II as detailed in the kit protocol. Cells were lysed, cleared by centrifugation, and supernatants were used for the assay. Lysates from the treated groups and controls were incubated with anti-LC3 antibody-coated microplates overnight. Plates were washed prior to exposure to detection and developing reagents. Absorbance was read at 450 nm.

### 2.7. Western Blotting Analysis for FoxO1, ATP1A1 and β1-Subunit of Na/K-ATPase (ATP1B1) Proteins

Cells/tissue lysates were homogenized in RIPA buffer (pH = 7.4) and centrifuged (14,000× *g* rpm/15 min/4 °C). Cell cytoplasmic/nuclear fractionations were performed using the NE-PER Nuclear and Cytoplasmic Extraction Reagents Kit (Thermo-Fisher Scientific, 78335, Waltham, MA, USA) and Nuclear Extraction Kit (Abcam, ab113474, Cambridge, MA). Supernatants were separated by SDS-PAGE and transferred to nitrocellulose membranes (Thermo Fisher Scientific, Waltham, MA, USA). Blocked membranes were incubated with primary antibodies (Appendix A), which was followed by their exposure to HRP-conjugated secondary antibodies. Proteins signals were detected using the Pierce ECL kit (Thermo Fisher Scientific, Waltham, MA, USA) and quantitated using ImageJ software (NIH, Bethesda, MD, USA) where the integrated density of each band was normalized against the housekeeping protein (β-Actin).

### 2.8. Protein Carbonylation and Reduced Glutathione (GSH)

The level of protein carbonylation in liver and tumor tissue was measured by ELISA (BioCell Protein Carbonyl Assay Kit; BioCell Corp., Auckland, New Zealand). Tissues were homogenized in 250 μL of ice-cold radioimmuno-precipitation (RIPA) assay buffer. Lysates were centrifuged at 16,000× *g* rpm for 15 min, and the supernatants were used for protein carbonylation analysis as per the kit instructions. A colorimetric GSH Assay Kit (Abcam Cambridge, MA, USA) was used to measure the GSH in liver tissue homogenates. Briefly, the tissue was rapidly homogenized in ice-cold 5% sulfosalicylic acid, and the supernatant was collected after 12,000× *g* rpm centrifugation at 4 °C for 20 min. After a 10-fold dilution in assay buffer, the GSH content in the supernatant was measured using the colorimetric micro-plate reader at OD450 and calculated based on the GSH standards.

### 2.9. RNA Isolation and Sequencing

The total RNA was extracted from Hep3B and SNU475 HCC cell lines using the RNeasy Mini Kit (Qiagen-74106, Germantown, MD, USA). A complete RNA-Seq analysis was performed by Novogene (Sacramento, CA, USA) once samples passed quality checks according to the criteria for RNA sequencing [29]. Briefly, a human-specific RNA library was prepared followed by library quality control. Illumina PE150 technology was employed for 150 bp paired-end sequencing of the samples. Finally, bioinformatic integration was performed to analyze the data. The KEGG pathways were specifically queried using the KEGG website (https://www.genome.jp/kegg/pathway.html), and selected KEGG pathway analyses were displayed utilizing the Pathview R package [30].

### 2.10. Statistical Analysis

Results are shown as box–whisker plots. Data are presented as median (central line), first and third quartiles (bottom and top of boxes, respectively), and whiskers (extreme values) from the indicated number of independent biological experiments. Differences among groups were determined by analysis of variance (ANOVA) followed by Tukey’s post hoc test or *t*-test, using GraphPad Prism version 9.0.1 (GraphPad, San Diego, CA, USA). The likelihood of less than 1/20 (*p* < 0.05) or 1/100 (*p* < 0.01) chance was considered statistically significant. Statistical tests, sample sizes and *p*-values are provided in the figure legends.

## 3. Results

### 3.1. H3K9ac and H3K9me3 in Human HCC Cell Lines

H3K9ac and H3K9me3 were significantly lower in histone extracts from human HCC cell lines treated with pNaKtide at 4 h and 18 h when compared to untreated cells (*p* < 0.01, Figure 1A). The effect of ATP1A1 signal normalization on H3K9ac and H3K9me3 levels in Hep3B and SNU475 cell lines was further confirmed by confocal immunofluorescence analysis (*p* < 0.01, Figure 1B).

### 3.2. ATP1A1 Expression in Human Normal Hepatocytes and Human HCC Cell Lines

We next evaluated the ATP1A1 expression in untreated and pNaKtide-treated human HCC cell lines. The immunofluorescence of ATP1A1 in human normal hepatocytes was not significantly different between pNaKtide-treated and untreated cells (Appendix A). In contrast, a significant up-regulation of ATP1A1 in the pNaKtide-treated HCC cell line was observed when compared to untreated cells (*p* < 0.01, Figure 2A,B). We further assessed the expression of other Na/K-ATPase subunit isoforms, including the β1 subunit (glycosylated and un-glycosylated forms). There was no significant difference in both glycosylated and un-glycosylated forms of ATP1B1 in human normal hepatocytes or in human HCC cell lines between pNaKtide treated vs. untreated cells by western blot or confocal microscopy (Figure 2C,D and Appendix A). Furthermore, no detectable expression of the ATP1A2 was noted in either human normal hepatocytes or human HCC cell lines (treated vs. untreated, Appendix A).

### 3.3. H3K9ac and H3K9me3 in the Xenograft SCID and MASH-HCC Mouse Models

Histone modifications were significantly lower in tumor tissue from pNaKtide treated vs. non treated SCID mice (*p* < 0.01, Figure 3A). Similarly, we observed a reduction in histone acetylation and tri-methylation by immunofluorescence (*p* < 0.01, Figure 3B). Furthermore, pNaKtide modulated cellular oxi-redox status as manifested by a significant decrease in protein carbonylation and concomitant increase in GSH levels (treated vs. untreated, *p* < 0.05, Figure 3C,D). These results supported the role of ATP1A1 in the regulation of ROI signaling, suggesting that the normalization of ATP1A1 signaling ameliorates abnormal epigenetic histone modifications in HCC.

The findings in the MASH-HCC mouse model concurred with the SCID mice model, where treated HFD mice with pNaKtide had a significantly lower intensity from labeled H3K9ac and H3K9me3 when compared to HFD mice at 24 and 28 weeks (low and high dose of pNaKtide, *p* < 0.05, *p* < 0.01, Figure 4A). In addition, hepatic protein carbonylation from treated mice significantly decreased when compared to non-treated animals (*p* < 0.05, *p* < 0.01, Figure 4B) with a concordant significant increase in GSH (treated vs. non-treated mice, *p* < 0.05, *p* < 0.01, Figure 4C). These results supported the role of ATP1A1 modulating ROI signaling exhausting endogenous antioxidant systems, which may in turn cause, at least in part, abnormal epigenetic histone modifications.

### 3.4. Cell Autophagy Activity in Human Normal Hepatocytes and HCC Cell Lines

Since oxidative signaling may play a major role in cell autophagy, the formation of autophagic vacuoles (autophagosomes) in normal and HCC cell lines was evaluated. Rapamycin and chloroquine were used as positive controls. Rapamycin is an autophagy inducer, and chloroquine induces the formation of autophagosomes in the early stages of autophagy. The visualization of autophagosomes in human normal hepatocytes by immunofluorescence showed no significant changes in pNaKtide-treated vs. untreated cells (Figure 5A). In contrast, exposure to pNaKtide, rapamycin, or chloroquine showed a significant increase in autophagic vacuoles in both human HCC cell lines compared to the untreated cells (*p* < 0.05, *p* < 0.01, Figure 5A). The immunofluorescent analysis of autophagy was confirmed quantitatively by measuring its surrogate marker LC3-II protein expression on human HCC cell line lysates. pNaKtide and chloroquine increased the level of LC3-II in Hep3B cell lines (Figure 5B). Furthermore, in SNU475 cell lines, LC3-II levels were increased in cells exposed to pNaKtide, chloroquine or rapamycin compared to untreated cells (*p* < 0.01, Figure 5B).

### 3.5. FoxO1 in Human Normal Hepatocytes and HCC Cell Lines

As the association between the transcriptional activation of a major tumor suppressor gene, FoxO3, and ATP1A1 signaling has been established [23], we proceed to explore the association of the ATP1A1 and the pro-autophagic gene, FoxO1 in MASH-HCC. Human normal hepatocytes treated with pNaKtide did not show any significant difference in nuclear FoxO1 expression compared to untreated cells (Appendix A). In contrast, both human HCC cell lines showed significant FoxO1 expression in the cell cytoplasm. This abnormal pattern was normalized upon cell lines exposure to pNaKtide, where treated cells showed a translocation of FoxO1 to the nucleus from their cytoplasm (*p* < 0.01, Figure 6A). Quantitation of the confocal images was confirmed by immunoblotting of FoxO1 in cell cytoplasm vs. nuclear fractions which showed a similar finding (nuclear untreated vs. nuclear treated, *p* < 0.01, Figure 6B, and cytoplasmic untreated vs. cytoplasmic treated, *p* < 0.05, *p* < 0.01, Figure 6B).

### 3.6. Gene Expression Profile and Pathways of HCC Progression in Human HCC Cell Line

mRNA sequencing analysis was performed to evaluate the differential gene expression in untreated vs. pNaKtide treated cells. The KEGG pathway maps, integrated using Pathview through queries to the KEGG database are displayed in Figure 7. Prominent metabolic pathways associated with HCC progression are demonstrated in the KEGG pathway analysis. As expected, a significant differential expression of genes involved in autophagy, mitophagy, FoxO family proteins, oxidative phosphorylation, TNF, NF-κB, and glutathione pathways were observed in pNaKtide-treated cells when compared to untreated cells (Figure 7). A list of major genes dysregulated in untreated Hep3B cells compared to pNaKtide treated Hep3B cells is detailed in Appendix A. However, further studies are warranted to explore the specific role of pNaKtide in regulating various genes involved in such pathways.

## 4. Discussion

HCC has a variety of biological behaviors associated with multiple histological phenotypes. This morphological multiplicity originates from not only disturbances on oncogenic and tumor suppressor pathways but also from genetic and epigenetic modifications [31,32]. Earlier reports have shown that the level of H3K9ac and H3K9me3 significantly correlates with tumor progression [33,34,35]. The present studies show that ATP1A1 signalosome dysregulation could lead to oxidative stress that mediates H3K9ac and H3K9me3 modifications and cell autophagy in HCC. Previous studies had shown that over-activation of the ATP1A1 pathway enhances an ROI amplification loop [20]. Such a progressive increase in ROI over time drives aberrant histone modifications and an inflammatory milieu, which affected cell autophagic activity, enhancing carcinogenesis [36]. We observed that the normalization of such a signalosome (ATP1A1) not only upregulated α1 subunit expression but restored cell autophagy activity concurrently with cell oxi-redox, i.e., decreased protein carbonyl and increased GSH levels, following a nuclear intrusion of cytoplasmic FoxO1.

In various pathological conditions, enhanced ROI production influences cell oxi-redox generating disturbances on ATP1A1 signaling, which are conditions that include a mitochondrial lipid overload observed in MASH [19,21,22,37]. ATP1A1 carbonylation by ROI leads to α-1/Src complex activation, i.e., Src → pSrc starting a cascade of Src kinase-mediated oxidant amplification loop [19,38]. Chronic oxidative stress causes damage to the hepatocyte, inducing cells to a senescence state that further intensifies inflammation and ROI production [39,40,41]. Oxidative stress can cause base and histone modifications, promoting genetic instability which increases the frequency of DNA mutations and aberrant genes expression involved in HCC development and progression [14,42]. Studies have shown a clear association between epigenetic instability and cell oxidative stress, and there is a mechanism for signaling inhibition as a potential therapeutic target [14,43,44]. In addition, protein carbonylation has been well established as a marker for the oxidant modulation of the ATP1A1 signaling function under oxidative stress [45,46]. In this study, a decreased level of protein carbonyls and increased level of GSH noted in pNaKtide-treated mice (normalization of the ATP1A1 signalosome) [23] were associated with a significant improvement in the antioxidant environment, which was conducive to the amelioration of cell epigenetic changes.

Former changes (H3K9ac and H3K9me3) have been correlated with tumor progression in multiple cancers [47,48,49]. Post-translational histone modification has been involved in transcriptional activation/inactivation, chromosome packaging, and DNA damage/repair, which coordinate various aspects of HCC development [5]. Hyperacetylation at the H3K9 of proto-oncogenes can cause nucleosome relaxation and the activation of tumorigenesis [50,51], since a loose chromatin structure serves as a marker of active transcription [52]. Trimethylation of histone H3 at lysine 9 plays a crucial role in cell migration through chromatin condensation, which can aggravate cancer progression [53,54,55] through an association with the silencing of tumor suppressor genes [56,57]. The present studies corroborate previous in vitro and in vivo reports where the dysfunctional epigenetic regulation of H3K9ac and H3K9me3 can exacerbate HCC progression [58,59,60,61,62,63,64]. Importantly, modulation of the ATP1A1 signal rescinded such histone modifications, portraying its pathway as a therapeutic target in HCC.

Cell autophagy activity may suppress tumor genesis by maintaining genomic stability and ensuring normal cell proteins function [17,65,66,67,68,69,70], while deranged autophagy may contribute to malignant transformation and tumor growth [71,72]. The removal of damaged mitochondria (mitophagy) may prevent ROI-driven tumor development by reducing oxidative stress [73,74]. In addition, epigenetic modifications regulate the complex process of autophagy and metabolic pathways in various cancer cells [75,76], enhancing a pro-tumorigenic inflammatory microenvironment [77]. The results of our study are in line with published reports [23], where a series of autophagy-related protein complexes were involved in the modulation of autophagosome formation. The major soluble microtubule-associated protein, light chain 3 (LC3), has been widely used as a potential marker of autophagy in a variety of tumor types including HCC [17,78,79,80,81]. The dysregulated cellular redox system impairs LC3-mediated autophagy through oxidation of the autophagy-related (ATG) conjugation systems [82]. During the process of autophagosome, the proteolytic cleavage of LC3 by the autophagy-related gene (Atg-4) leads to the formation of LC3-I, which in turn conjugates to phosphatidylethanolamine generating LC3-II [83,84], and LC3-II is a marker for the successful formation of the autophagosome [85]. Our results showed a significant reduction in LC3 in human HCC. Interestingly, pNaKtide significantly increased LC3-II in two human HCC lines, which is perhaps due to its potent ROI regulation from normalizing ATP1A1 signaling.

FoxO1, the first described of the FoxO transcription factors, improves metabolism not only by enhancing cell insulin metabolic actions but also by promoting cell autophagy activity [86,87,88,89,90]. FoxO-proteins effectively prevent the carcinogenic effects of ROI and act as an important defense against cellular oxidative stress in various cancers, including MASH-HCC [87,91,92,93,94,95]. Our results reconciliate with published reports, and normalization of the ATP1A1 significantly up-regulated nuclear FoxO1 protein in HCC cell lines. In addition, we tested for cell autophagy activity and FoxO1 expressions in normal hepatocytes and HCC cell lines. Since liver homogenates would include a variety of malignant and benign hepatocytes plus non-parenchymal cells, an assay on autophagy activity in vivo may not reflect the malignant cell profile. Future studies will be designed to eliminate the limitation of the present study on FoxO1-mediated cell autophagy from the HCC–xenograft and MASH-HCC models. RNA sequencing on human HCC cell lines supports our results and reinforces the prominent role of the ATP1A1 signalosome in cell oxidation that upon further activation promotes epigenetic modifications in HCC with a decline on cell autophagy activity. Further mechanistic studies are needed to fully explore the role of ATPIA1 signaling complex in epigenetic–autophagic-mediated hepatocarcinogenesis. The bioavailability studies of pNaKtide are essential to confirm the organ-specific distribution of the peptide. The underlying mechanisms behind the role of the ATP1A1 signalosome in the epigenetic modifications and cell autophagy activity during HCC need to be further determined.

## 5. Conclusions

The progression from MASH to HCC is enhanced by cell oxidative stress and the inflammatory microenvironment, which are known to cause aberrant epigenetic modifications that trigger HCC progression. The activation of the ROI amplification loop at the ATP1A1 signalosome favored tumor progression by enhancing epigenetic instability, creating a paucity of cell autophagic activity. Normalization of ATP1A1 signaling modulated ROI, played a causative role in ameliorating epigenetic histone modifications and restored cell autophagy activity, ultimately decreasing HCC development and progression.

## Figures and Tables

**Figure 1 cells-12-02367-f001:**
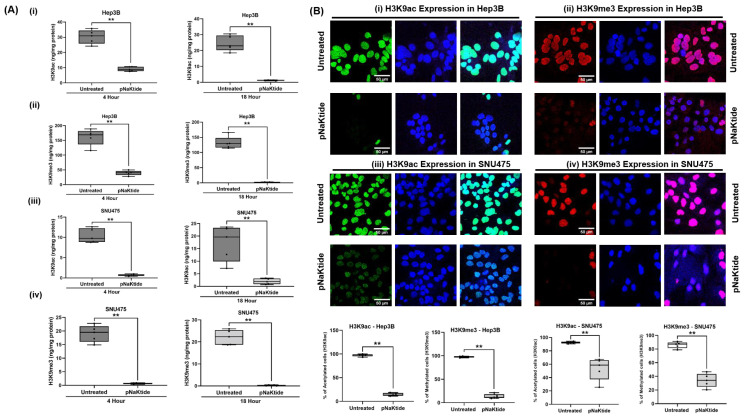
Epigenetic histone modifications in two human HCC cell lines. Incubation with pNaKtide (4 h and 18 h, at a concentration of 62.5 and 6 µM for Hep3B and SNU475 human HCC cell lines, respectively) significantly decreased the level of H3K9ac and H3K9me3 in HCC cell lines. (**A**) Quantitative analysis of H3K9ac and H3K9me3 in Hep3B (**i** and **ii**) and SNU475 (**iii** and **iv**) cell lines by ELISA. (*n* = 5, results shown as box–whisker plots. ** *p* < 0.01 by *t*-test). (**B**) Confocal microscopy images and quantification of immunofluorescence staining of H3K9ac and H3K9me3 in Hep3B (**i** and **ii**) and SNU475 (**iii** and **iv**) cell lines exposed to treated and untreated for 4 h (*n* = 5, results shown as box–whisker plots. ** *p* < 0.01 by *t*-test). Scale bar = 50 µm. Green, red, and blue fluorescence indicates the expressions of H3 acetyl K9, H3 tri-methyl K9 and DAPI, respectively.

**Figure 2 cells-12-02367-f002:**
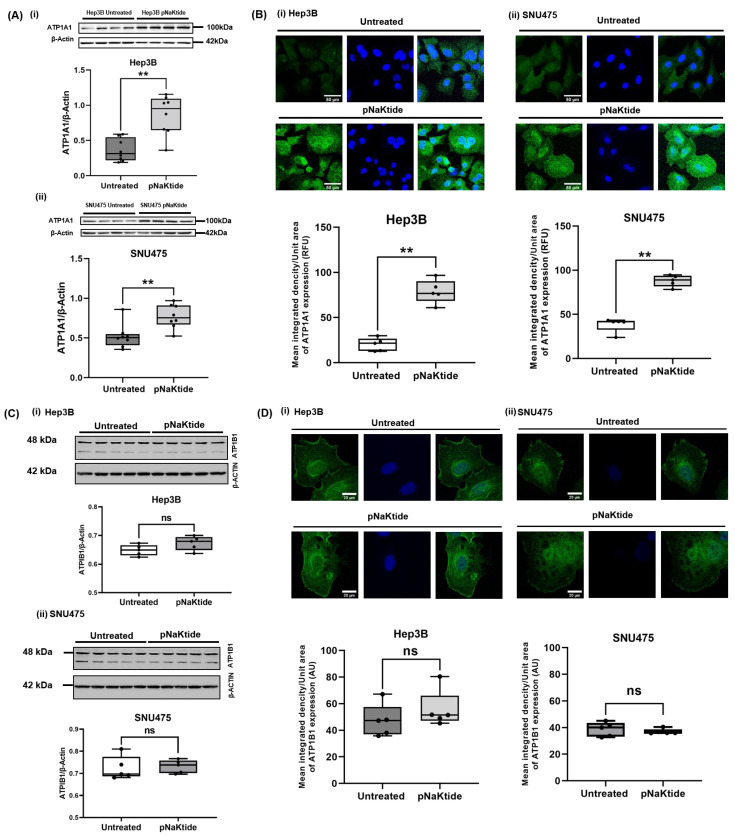
Expression of ATP1A1 and ATP1B1 subunits in two human HCC cell lines. pNaKtide treatment (4 h, at a concentration of 62.5 and 6 µM for Hep3B and SNU475 human HCC cell lines, respectively) significantly up-regulated the expression of ATP1A1 in HCC cell lines, while no significant changes were observed in the expression of ATP1B1 compared to the untreated group. (**A**) Immunoblot of ATP1A1 in (**i**) Hep3B and (**ii**) SNU475 cell lines (*n* = 8, results are shown as box–whisker plots. ** *p* < 0.01 by *t*-test). (**B**) Confocal microscopy images showing ATP1A1 expressions in (**i**) Hep3B and (**ii**) SNU475 cell lines. (*n* = 5, results are shown as box–whisker plots. ** *p* < 0.01 by *t*-test). (**C**) Immunoblot of ATP1B1 in (**i**) Hep3B and (**ii**) SNU475 cell lines (*n* = 5, results are shown as box-whisker plots. No significant difference (ns) by *t*-test). (**D**) Confocal microscopy images and quantification showing immunofluorescence staining of ATP1B1 in (**i**) Hep3B and (**ii**) SNU475 cell lines. (*n* = 5, results are shown as box–whisker plots. No significant difference (*ns*) by *t*-test). Scale bar = 50 µm.

**Figure 3 cells-12-02367-f003:**
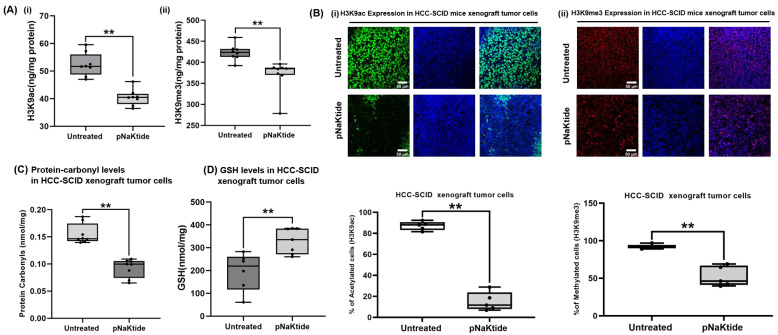
Epigenetic histone modifications and oxidative stress in the HCC-SCID tumor xenograft mouse model. pNaKtide administration (30 mg/kg of BW, three times per week) significantly decreased the levels of H3K9ac and H3K9me3 in tumor xenografts of SCID mice; (**A**) Quantitative analysis of (**i**) H3K9ac and (**ii**) H3K9me3 in tumor xenografts of SCID mice by ELISA (*n* = 8, results are shown as box–whisker plots. ** *p* < 0.01 by *t*-test). (**B**) Confocal microscopy images of (**i**) H3K9ac and (**ii**) H3K9me3 in the HCC-SCID mouse tumors. (*n* = 5, results are shown as box–whisker plots. ** *p* < 0.01 by *t*-test). Scale bar = 50 µm. (**C**) pNaKtide administration (30 mg/kg of BW, three times per week) significantly decreased the level of protein carbonyls in the tumor xenograft of SCID mice, whereas GSH concentration was significantly increased compared to untreated animals. Protein carbonylation level in the tumor xenografts from SCID mice by ELISA (*n* = 8, results are shown as box–whisker plots. ** *p* < 0.01 by *t*-test). (**D**) GSH concentration in the tumor xenograft of SCID mice (*n* = 6–7, results are shown as box–whisker plots. ** *p* < 0.01 by *t*-test).

**Figure 4 cells-12-02367-f004:**
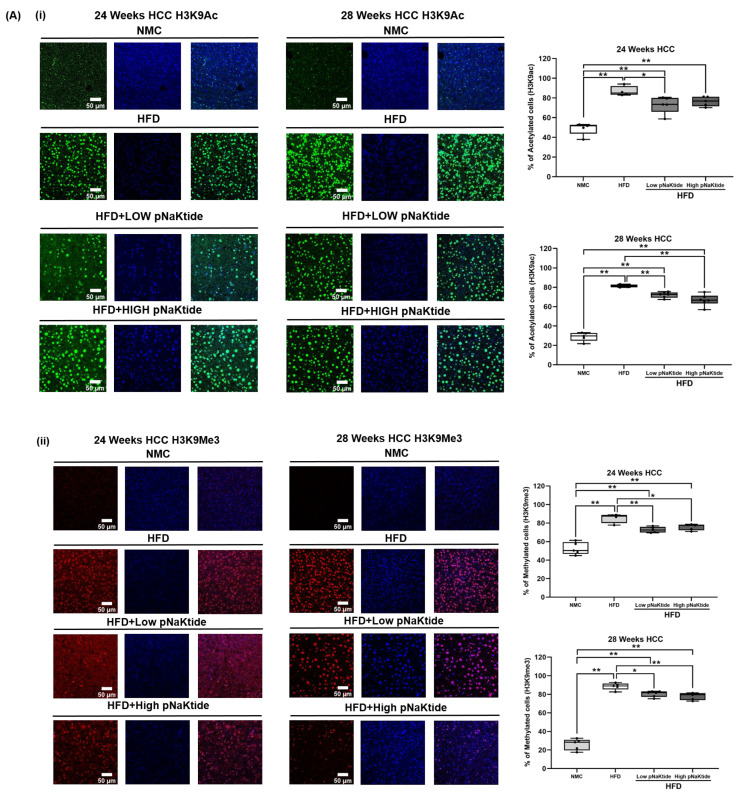
Epigenetic histone modifications and oxidative stress in the MASH-HCC mouse model. pNaKtide administration (2 mg/kg or 10 mg/kg of BWX3 a week) significantly decreased the level of H3K9ac and H3K9me3 and protein carbonyls in the liver from treated vs. untreated animals at 24 and 28 weeks of MASH-HCC mouse, whereas GSH concentration was significantly increased in the livers from pNaKtide-treated MASH-HCC mice. (**A**) Confocal microscopy images of (**i**) H3K9ac and (**ii**) H3K9me3 in the liver of MASH-HCC mice (24 and 28 weeks of treatment, *n* = 5, results are shown as box–whisker plots. * *p* < 0.05, ** *p* < 0.01 by ANOVA and Tukey’s post hoc test; Scale bar = 50 µm). (**B**) Protein carbonylation level in the MASH-HCC mouse liver by ELISA (24 and 28 weeks of treatment *n* = 4–6, results are shown as box–whisker plots. * *p* < 0.05, by ANOVA and Tukey’s post hoc test). (**C**) GSH concentration in the MASH-HCC mouse liver (24 and 28 weeks of treatment, *n* = 4–6, results are shown as box–whisker plots. * *p* < 0.05, ** *p* < 0.01 by ANOVA and Tukey’s post hoc test).

**Figure 5 cells-12-02367-f005:**
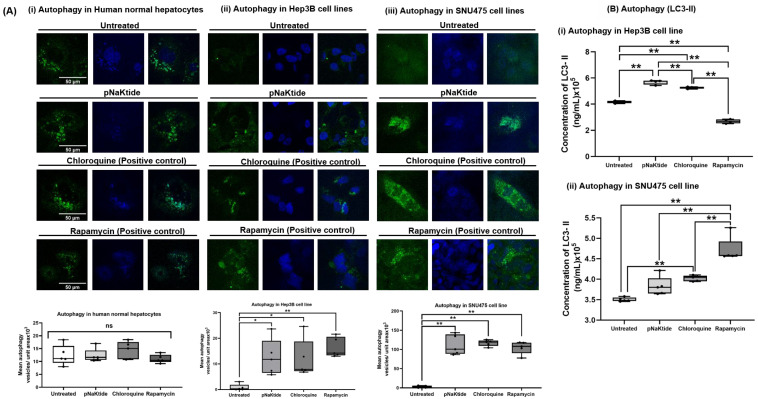
Autophagy in human normal hepatocytes and two human HCC cell lines. pNaKtide treatment (4 h at a concentration of 10, 62.5 and 6 µM for human normal hepatocytes, Hep3B and SNU475 human HCC cell lines, respectively) increased autophagic activity in both HCC cell lines compared to untreated HCC cell lines (*p* < 0.05), while no significant changes (ns) were observed in human normal hepatocytes. (**A**) Detection of immunofluorescence of autophagic vacuoles in (**i**) human normal hepatocytes but a paucity of immunofluorescence in untreated HCC cell lines (**ii**) Hep3B and (**iii**) SNU475 (*n* = 5, results are shown as box–whisker plots. * *p* < 0.05, ** *p* < 0.01 by ANOVA and Tukey’s post hoc test). Scale bar = 50 µm. (**B**) Quantitation of LC3-II in HCC cell lines ((**i**). Hep3B and (**ii**). SNU475) by ELISA (*n* = 5, results are shown as box-whisker plots. ** *p* < 0.01 by ANOVA and Tukey’s post hoc test).

**Figure 6 cells-12-02367-f006:**
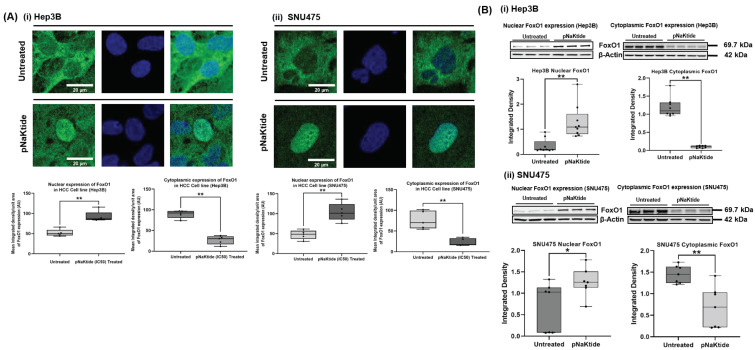
Normalization of the ATP1A1 effect on the nuclear translocation of FoxO1 in human HCC cell lines. Nuclear translocation of FoxO1 was significantly increased in pNaKtide-treated group compared to untreated HCC cell lines. (**A**) Confocal microscopy images and quantification showing immunofluorescence staining of FoxO1 in HCC cell lines (**i**) Hep3B and (**ii**) SNU475 (*n* = 5, images of cells were taken at 63× magnification with a zoom factor of 5.0, results are shown as box–whisker plots. ** *p* < 0.01 by *t*-test). Scale bar = 20 µm. (**B**) Immunoblot of FoxO1 in nuclear and cytoplasmic fractions of HCC cell lines (**i**) Hep3B and (**ii**) SNU475 (*n* = 7–9, images of cells were taken at 63× magnification with a zoom factor of 5.0, results are shown as box-whisker plots. * *p* < 0.05, ** *p* < 0.01 by *t*-test).

**Figure 7 cells-12-02367-f007:**
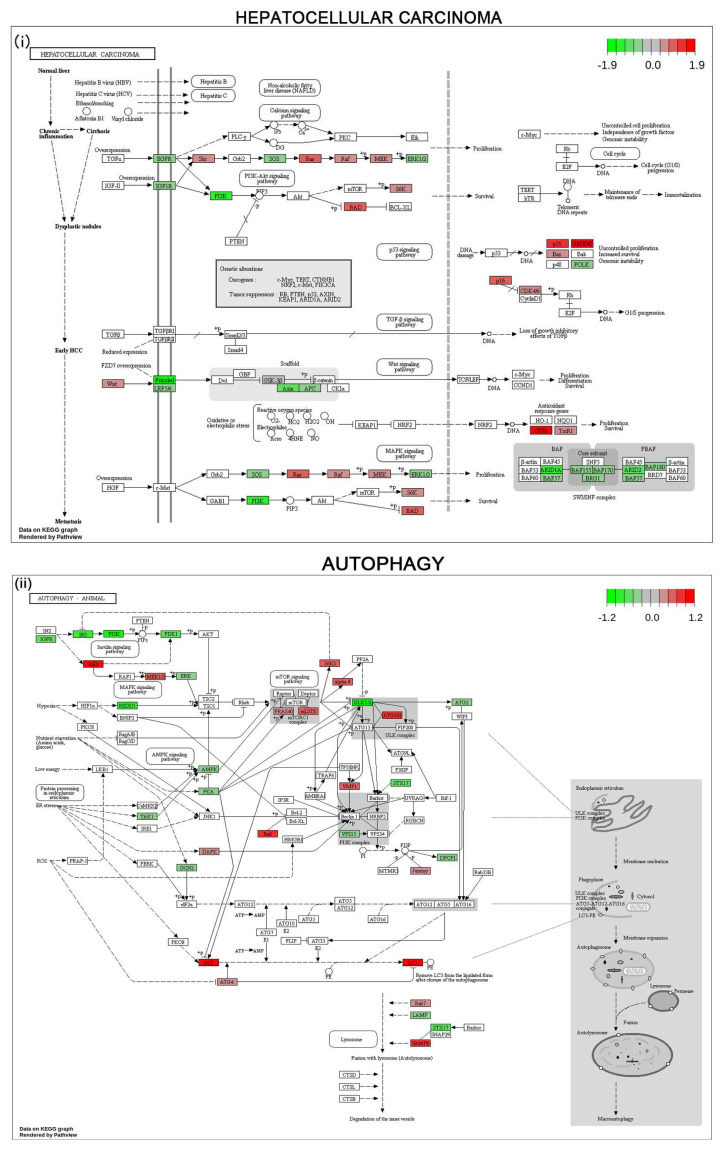
Molecular pathways in Hep3B cell line. KEGG pathway diagrams show the enrichment of metabolic and signaling pathways in treated vs. untreated HCC cancer cell line (Hep3B). Relative gene expressions derived from RNAseq analyses are depicted in pathways with red indicating up- and green indicating downregulation, respectively. (**i**) KEGG pathway for hepatocellular carcinoma. (**ii**) KEGG pathway for autophagy. (**iii**) KEGG pathway for pathways in cancer. (**iv**) KEGG pathway for mitophagy. (**v**) KEGG pathway for oxidative phosphorylation. (**vi**) KEGG pathway for TNF signaling pathway. (**vii**) KEGG pathway for NF-Kappa B signaling pathway. (**viii**) KEGG pathway for glutathione metabolism.

## Data Availability

Data available upon request from the corresponding author.

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
