# Peer review of "Normalization of the ATP1A1 Signalosome Rescinds Epigenetic Modifications and Induces Cell Autophagy in Hepatocellular Carcinoma†"

_cells, 2023, doi:10.3390/cells12192367_

Round 1

Reviewer 1 Report

In this manuscript, the authors reported that inhibition of ATP1A1 signaling induces epigenetic changes and autophagy in MASH-HCC and xenograft models, thereby suppressing HCC tumor growth.

Major points:

1.     While the authors observed epigenetic alterations upon pNaKtide treatment, whether these alterations contribute to tumor suppressive function is unclear. What is the significance of these changes? There is no evidence to show that these changes indeed cause tumor suppression.

2.     Did pNaKtide treatment suppress tumor growth in xenograft model?

3.     What are the results of MASH-HCC model? Tumor number and size? Histology? fibrosis? lipid accumulation? MASH severity?

4.     The authors used ELISA quantification of LC3-II as a marker of autophagy. However,LC3-II/LC3-I ratio is a more accurate autophagy marker. Western blot of LC3-II/LC3-I ration is needed.

5.     In Fig. S1, and 4A, quantification with immunofluorescence is not accurate. Western blot analysis is needed for quantification.

6.     Signals in FIg. S2 look like nonspecific. If p<0.01, why the authors concluded that there is no significant difference in nuclear FoxO1 between pNaKtide treated and untreated cells?

7.     List of dysregulated genes should be included in the paper as a supplemental table.

Minor points:

No Figure S7 was included in the manuscript. Uncropped western blot data were included as figure S11 and S12, but not mentioned in the text.  

Author Response

Reviewer 1

1.While the authors observed epigenetic alterations upon pNaKtide treatment, whether these alterations contribute to tumor suppressive function is unclear. What is the significance of these changes? There is no evidence to show that these changes indeed cause tumor suppression,

  1. Did pNaKtide treatment suppress tumor growth in the xenograft model?
  2. What are the results of MASH-HCC model? Tumor number and size? Histology? fibrosis? lipid accumulation? MASH severity?

Ans: Our present work builds in our prior publication that demonstrated normalization of the ATP1A1 signalosome portrays as a tumor suppressor for MASH-HCC through an apoptotic switch on malignant cells, significantly decreasing tumor burden in-vitro (human HCC cell line xenograft transplanted into the SCID mice model) and in-vivo (MASH-HCC mice model) (PMID: 35806364). Cellular-redox imbalance and chronic inflammation are observed when the α1-subunit of Na/K-ATPase (ATP1A1) signaling is deregulated as reported in various pathological conditions [PMID: 33356956, PMID: 30041449, PMID: 33275652, PMID: 17296611, PMID: 27445847]. Hence in the present study, we aimed to explore the role of aberrant epigenetic modification mediated through Na-K ATPase-mediated oxidative signaling pathways in the progression of HCC and the possible amelioration by pNaKtide. The results of the present study show that pNaKtide could significantly modulate the altered epigenetic modification through the normalization of ATP1A1 signaling by mitochondrial renewal and restoration of metabolic function.

  1. The authors used ELISA quantification of LC3-II as a marker of autophagy. However, LC3- II/LC3-I ratio is a more accurate autophagy marker. Western blot of LC3-II/LC3-I ratio is needed.

Ans: We agree with the reviewer's suggestion. We have quantitated autophagy by ELISA and confocal microscopy with a similar trend. Unfortunately, due to limited tissue availability in our animal models, we couldn’t include western blotting as a marker of autophagy. As suggested by the reviewer, we will consider LC3-II/LC3-I ratio in our future studies.

  1. In Fig. S1, and 4A, quantification with immunofluorescence is not accurate. Western blot analysis is needed for quantification.

Ans: We have included, for the epigenetic change’s quantitation, confocal (provides a visual effect) and ELISA methods. Both methods reconciled well in their trend by group and treatment. We believe they provide accurate, reliable, and reproducible results. Nevertheless, and due to limited tissue availability, we have not included blotting techniques. We will consider the western blot analysis in future studies. In Fig. S1 (A & B), we have shown the effect of pNaKtide on ATP1A1 and ATP1B1 in normal hepatocytes. pNaKtide had no effect on ATP1A1 signaling pathway. Fig. S1C shows no significant expression of the α-2 subunit of the Na/K-ATPase in both normal hepatocytes and HCC cell lines. We have included these findings as supporting data to our main findings. We will consider the western blot analysis in future studies.

  1. Signals in Fig. S2 look nonspecific. If p<0.01, why the authors concluded that there is no significant difference in nuclear FoxO1 between pNaKtide treated and untreated cells?

Ans: Thanks for the careful observation. There was a typological error in the results section of Fig S2. We have removed p<0.01 from the results.

  1. List of dysregulated genes should be included in the paper as a supplemental table.

Ans: As suggested by the reviewer, we have included the list of dysregulated genes in supplemental Table 2.

Minor points:

No Figure S7 was included in the manuscript. Uncropped western blot data were included as figure S11 and S12, but not mentioned in the text.

Figure S7 from our previous study was cited for the animal flow chart of the study. The sentence has been modified accordingly to avoid confusion.

Uncropped western blot data has been included as additional raw data.

Reviewer 2 Report

Pradeep Kumar Rajan reviewed the effect of α1-subunit of the Na/K-ATPase (ATP1A1) signalosome normalization on post-translational histone modifications and cell autophagy and their impact in MASH-associated HCC development and progression. They found that ATP1A1 signalosome dysregulation activated the ROI amplification loop, favoured tumour progression via epigenetic modifications and impaired cell autophagy activity. I believe the results are of interest. However, there are several suggestions need to be addressed before publication.

Major revisions:

1. In this paper, image resolution of all figures was too low to difficult to identify the results, please change them. 

2. In this study, the authors demonstrated that ATP1A1 signalosome regulates ROI signaling, rescinds epigenetic modifications, and induces cell autophagy. However, the relationship between ROI signaling and epigenetic modifications was not further determined, so the conclusion that Further activation of the ROI amplification loop at the ATP1A1 signalosome favoured tumour progression by enhancing epigenetic instability creating a paucity of cell autophagic activity was not suitable in the part of 5. Discussion.  Minor comments:

1. In line 155 and 157, what is the meaning of ºC, please confirm them.

2. In Figure 2, ATP1B1 and ATP1A2 were detected; the authors should discuss the aim of this experiment.

3. The results of RNA-sequence should be added in the text, rather than the supplementary file(s).

Author Response

Dear Editorial Team and reviewers,

Thank you for your time and effort in reviewing our manuscript, “Normalization of the ATP1A1 Signalosome Rescinds Epigenetic Modifications and Induces Cell Autophagy in MASH-Related Hepatocellular Carcinoma” cells-2579243. We are very pleased with your kind response towards improving our work. In the following pages, we detail our responses to the reviewers’ comments and the changes/additions have been included in the manuscript as track changes.

Reviewer 2.  

Thank you for the comments.

  1. In this paper, image resolution of all figures was too low to difficult to identify the results, Please, change them.

Ans: We have uploaded the high-resolution images along with the manuscript file.

  1. In this study, the authors demonstrated that ATP1A1 signalosome regulates ROI signaling, rescinds epigenetic modifications, and induces cell autophagy. However, the relationship between ROI signaling and epigenetic modifications was not further determined, so the conclusion that further activation of the ROI amplification loop at the ATP1A1 signalosome favored tumor progression by enhancing epigenetic instability creating a paucity of cell autophagic activity was not suitable in the part of the Discussion.

Ans: Previous studies have already demonstrated that reactive oxygen species and oxidative stress signaling pathways have pivotal roles in various epigenetic modifications that promote tumor progression in HCC (PMID: 31056726, PMID: 35806364, PMID: 24281019). Furthermore, the activation of ROI signaling is associated with deregulated Na/K-ATPase signaling in various pathological conditions [PMID: 33356956, PMID: 30041449, PMID: 33275652, PMID: 17296611, PMID: 27445847] and our previous study has demonstrated the tumor suppressor role of pNaKtide mediated through the blockage of Na-K ATPase-mediated ROI amplification (PMID: 35806364). Hence in the present study, we aimed to explore the role of aberrant epigenetic modification and autophagy mediated through Na-K ATPase-favored ROI in the progression of HCC and the possible amelioration by pNaKtide. As per the reviewer’s suggestion, the conclusion has been revised accordingly.

Minor comments.

  1. In line 155 and 157, what is the meaning of ºC, please confirm them.

4 Degree Celsius (4°C) and -80 Degree Celsius (-80°C)

  1. In Figure 2, ATP1B1 and ATP1A2 were detected; the authors should discuss the aim of this experiment.

Ans: The aim of the present study was to explore the specific role of α1 subunit of Na/K-ATPase (ATP1A1) signaling in ROI amplification and the resulting alteration of epigenetic pathways and autophagy. We have included the results of other sub-units of Na/K-ATPase viz, ATP1B1 and ATP1A2 to demonstrate the sub-unit-specific role of Na/K-ATPase in ROI amplification and no significant changes were observed in the expression of either ATP1B1 or ATP1A2.

  1. The results of RNA-sequence should be added in the text, rather than the supplementary file(s).

Ans: As per the reviewer’s suggestion, the results of the RNA-sequencing experiments have been included in the main manuscript as Fig. 7.

Round 2

Reviewer 1 Report

While I am not satisfied with the responses from the authors, I understand the limited availability of tissue samples that prevented the authors to perform suggested experiments. 

Author Response

Thank you so much for the suggestions. We will consider the suggested western blot experiment in our future studies.